# Critical Review of the Methodological Shortcoming of Ambulatory Blood Pressure Monitoring and Cognitive Function Studies

**DOI:** 10.3390/clockssleep7010011

**Published:** 2025-03-06

**Authors:** Shahab Haghayegh, Ramon C. Hermida, Michael H. Smolensky, Mili Jimenez Gallardo, Claudia Duran-Aniotz, Andrea Slachevsky, Maria Isabel Behrens, David Aguillon, Hernando Santamaria-Garcia, Adolfo M. García, Diana Matallana, Agustín Ibáñez, Kun Hu

**Affiliations:** 1Department of Anesthesia, Critical Care and Pain Medicine, Massachusetts General Hospital, Boston, MA 02114, USA; mjimenezgallardo@mgh.harvard.edu (M.J.G.); khu1@mgh.harvard.edu (K.H.); 2Harvard Medical School, Boston, MA 02115, USA; 3Broad Institute, Cambridge, MA 02142, USA; 4Bioengineering & Chronobiology Laboratories, Atlantic Research Center for Telecommunication Technologies (atlanTTic), Universidade de Vigo, 36310 Vigo, Spain; rhermida@uvigo.es; 5Bioengineering & Chronobiology Research Group, Galicia Sur Health Research Institute (IIS Galicia Sur), SERGAS-UVIGO, 36213 Vigo, Spain; 6Department of Biomedical Engineering, Cockrell School of Engineering, The University of Texas at Austin, Austin, TX 78705, USA; michael.h.smolensky@uth.tmc.edu; 7Department of Internal Medicine, Division of Cardiology, McGovern School of Medicine, The University of Texas Health Science Center at Houston, Houston, TX 77030, USA; 8Latin American Brain Health Institute (BrainLat), Universidad Adolfo Ibanez, Santiago 7941169, Chile; duran.aniotz@gmail.com (C.D.-A.)agustin.ibanez@gbhi.org (A.I.); 9Gerosciences Center for Brain Health and Metabolism (GERO), Santiago 7500922, Chile; andrea.slachevsky@uchile.cl (A.S.); hernando.santamaria@javeriana.edu.co (H.S.-G.); 10Memory and Neuropsychiatric Center (CMYN) Neurology Department, Hospital del Salvador & Faculty of Medicine, University of Chile, Santiago 8380453, Chile; 11Neuropsychology and Clinical Neuroscience Laboratory (LANNEC), Physiopathology Department—Institute of Biomedical Science, Neuroscience and East Neuroscience Departments, Faculty of Medicine, University of Chile, Santiago 8380453, Chile; 12Neurology and Psychiatry Department, Clínica Alemana-University Desarrollo, Santiago 7650567, Chile; behrensl@uchile.cl; 13Centro de Investigación Clínica Avanzada (CICA), Universidad de Chile, Santiago 8380453, Chile; 14Departamento de Neurología y Neurocirugía, Hospital Clínico de la Universidad de Chile, Santiago 8380453, Chile; 15Departamento de Neurociencia, Universidad de Chile, Santiago 8380453, Chile; 16Grupo de Neurociencias de Antioquia, Facultad de Medicina, Universidad de Antioquia, Medellin 050010, Colombia; david.aguillon@gna.org.co; 17Cognitive Neuroscience Center, Universidad de San Andrés, Buenos Aires B1644BID, Argentina; amgarcia@udesa.edu.ar; 18Global Brain Health Institute (GBHI), University of California San Francisco, San Francisco, CA 94143, USA; 19Trinity College Dublin, D02 X9W9 Dublin, Ireland; 20Departamento de Lingüística y Literatura, Facultad de Humanidades, Universidad de Santiago de Chile, Santiago 8380453, Chile; 21Neuroscience Program, Aging Institute, Psychiatry Department, School of Medicine, Pontificia Universidad Javeriana, Bogotá 110231, Colombia; dianamat@javeriana.edu.co; 22Mental Health Department, Hospital Universitario Fundación Santa Fe, School of Medicine, Universidad de los Andes Bogota, Bogotá 110111, Colombia; 23Department of Medicine, Brigham and Women’s Hospital, Boston, MA 02115, USA

**Keywords:** ambulatory blood pressure monitoring, dementia, hypertension, cognitive decline

## Abstract

Growing evidence suggests that abnormal diurnal blood pressure rhythms may be associated with many adverse health outcomes, including increased risk of cognitive impairment and dementia. This study evaluates methodological aspects of research on bidirectional associations between ambulatory blood pressure monitoring (ABPM) patterns and cognitive function. By examining the 28 recent studies included in a recent systematic review on the association between ABPM patterns with cognitive function and risk of dementia, our review revealed several significant limitations in study design, sample characteristics, ABPM protocol, cognitive assessment, and data analysis. The major concerns include a lack of diversity in study populations with underrepresentation of Blacks and Latinos, a predominant focus on Alzheimer’s disease or all-cause dementia without distinguishing other dementia subtypes, different and not standardized measures of cognition or dementia, prevalent use of 24 h monitoring without considering the adaption effect, inconsistent definitions of dipping status, and ignorance of individual differences in timings of daily activities such as bed and awakening times. In addition, confounding variables such as class, dose, and timing of antihypertensive medication are inadequately controlled or considered. Further, longitudinal studies were scarce examining the bidirectional relationship between ABPM patterns and cognitive decline over time. Collectively, these deficiencies undermine the reliability and generalizability of current findings. Addressing these methodological challenges is crucial for a more comprehensive understanding of diurnal blood pressure rhythms in diverse populations and for developing an evidence-based guideline for ambulatory monitoring and control of blood pressure across the sleep-wake cycle to prevent cognitive decline and dementia.

## 1. Introduction

Hypertension is a major risk factor for cognitive decline and dementia. The relationship between blood pressure (BP) and cognitive function is complex and bidirectional. Chronic hypertension can lead to cerebrovascular damage, disruption of the blood-brain barrier, and alterations of cerebral blood flow autoregulation, all of which can contribute to cognitive impairment [1]. Conversely, cognitive decline and dementia can affect BP control through changes in autonomic regulation and also medication adherence [2].

Compared to conventional single daytime office BP measurement (OBPM), around-the-clock ambulatory BP monitoring (ABPM) provides a more comprehensive assessment of BP patterning throughout the 24 h, capturing diurnal variations and nocturnal dipping status [3]. In normal, healthy, day-active persons, systolic and diastolic blood pressure (SBP, DBP) vary in a rather predictable-in-time fashion during the daily cycle of activity and rest. They are usually lowest during mid-sleep (dipping), rise before awakening and progressively continue to do so until midday/early afternoon when they undergo minor decline, and then increase to peak values in the late afternoon/early evening before again declining to lowest levels during sleep [4]. Growing evidence suggests that non-dipping and reverse dipping BP patterns are associated with increased risk for cognitive impairment and dementia [5]. The mechanisms underlying this association likely involve chronic cerebral hypoperfusion, increased blood-brain barrier permeability, and accelerated neurodegeneration [6].

In recent years, there has been growing interest in understanding the complex relationship between diurnal BP patterning and cognitive function [7,8,9,10,11,12]. A recent systematic review and meta-analysis conducted by Gavriilaki et al. [5] examined the relationship between features of ABPM-derived BP patterning and cognitive function across 28 studies involving 7595 participants. Their meta-analysis focused on synthesizing outcome data, reporting that individuals exhibiting normal nocturnal BP dipping had a 51% lower risk of cognitive impairment or dementia compared to non-dippers, and reverse dippers had up to a 6-fold higher risk of cognitive impairment compared to dippers. While their work assessed the strength of these associations, questions remain about the methodological rigor of the evidence supporting these relationships. Our present analysis takes a fundamentally different approach by conducting an in-depth examination of the research methods used in these studies, aiming to identify key methodological limitations that could affect the reliability and generalizability of findings in this field.

The need for such methodological evaluation is clear, as variability in protocol design, definition of dipping status, cognitive assessment, and control for confounding factors across studies introduces heterogeneity that may hinder the application of ABPM in health care and clinical practice. Additionally, the predominance of cross-sectional vs. longitudinal protocols limits causal inference of the relationship between BP patterning and cognitive outcomes. This methodological focus is crucial for improving future research design and ultimately strengthening our understanding of how BP patterns relate to cognitive outcomes.

Herein, we evaluate the methods of those studies included in the systematic review by Gavriilaki et al. [5] We aim to identify critical limitations and factors that may affect the quantification of the bidirectional association between 24 h BP rhythms and cognitive function. By highlighting the shortcomings of past investigations, we intend to inform more rigorous study designs that more reliably elucidate the complex reciprocal relationship between BP dysregulation and cognitive decline. 

## 2. Review Methodology and Data Extraction

We systematically evaluated the methods utilized in each of the 28 investigations comprising the systematic review and meta-analysis by Gavriilaki et al. (2023) [5] concerning the association between ABPM-derived BP patterning and cognitive function or risk of dementia. These original studies, identified through a comprehensive search of PubMed, Embase, and Cochrane databases, involved studies of ≥10 participants (each study) that reported on all-cause dementia, cognitive impairment—based on validated cognitive tests, and features of ABPM-derived 24 h BP patterning. Investigations were excluded if they examined the effect of an intervention on BP dipping and cognitive function. 

To evaluate the methodological aspects of these studies, we extracted relevant information from each of the reviewed articles using a pre-determined list of relevant elements to capture key aspects of study design and execution that could impact the validity and reliability of findings. The pre-determined list of extracted data/information encompassed study design, sample size, subject characteristics, duration of ABPM, frequency of BP measurements, definition of BP dipping patterns, criteria for determining validity of ABPM measurements, method of determining daytime wake and nighttime sleep periods, inclusion of follow-up ABPM and cognitive assessments (where applicable), report of sample size/effect size calculation, statement of participant dropout rate, control for confounding factors, and listing or control of the timing of BP medication. Initial data extraction was performed using Claude AI (Claude 3.5 Sonnet, developed by Anthropic, San Francisco, California, USA), an artificial intelligence language model. Two of the authors (S.H., M.J.G.) thoroughly and separately reviewed and revised the extracted information to ensure its accuracy and completeness.

## 3. Assessment of Methodological Characteristics

Our comprehensive analysis of the 28 studies revealed significant heterogeneity of their investigative methods (Table 1 and Appendix A). Most studies [7,8,10,12,13,14,15,16,17,18,19,20,21,22,23,24,25,26,27,28,29,30,31,32,33] (25/28, 89.3%) were cross-sectional in design, and only a small proportion (3/28, 10.7%) were longitudinal [9,11,34]. There were considerable variations in sample size (range: 30–1608; median: 174) and age of participants (mean age range: 54.3–93.2 years). Gender distribution varied significantly across studies, with male participation ranging from 10% to 100% (median: 53.7%). There was limited reporting of demographic information, with race/ethnicity unreported in 82% (23/28) of studies. Among studies that did report racial/ethnic demographics, only two of them [9,30] included Black or Latino participants.

Methods of cognitive assessments varied between investigations, with the Mini-Mental State Examination (MMSE) most frequently utilized (16/28, 57.1%) [10,12,14,15,16,20,22,23,24,25,27,29,30,32,33]. Other common cognitive assessments included the Montreal Cognitive Assessment (MoCA) [16,21,33] (3/28, 10.7%) and Trail Making Test [16,25,30] (3/28, 10.7%) among various other neuropsychological tests. The definition of cognitive impairment and dementia lacked standardization across studies, with criteria ranging from specific cut-off scores on cognitive tests to clinical diagnoses based on established criteria. The majority of studies did not specify dementia subtypes, instead focusing on extent cognitive impairment, e.g., mild cognitive impairment, using general cognitive assessments. Alzheimer’s Disease (AD) was specifically investigated in only four studies [8,10,13,23] (14.3%), and vascular dementia was examined in two studies [8,19] (7.1%), while other studies focused on cognitive impairment without specifying dementia subtypes.

All studies (28/28, 100%) conducted ABPM solely for 24 h, with BP typically sampled every 15–30 min during the daytime hours and every 30–60 min during the nighttime hours. Data quality control measures for ABPM measurements varied widely, ranging from stringent criteria to unreported procedures (18 or 64.3% of studies [7,8,10,12,13,15,16,18,19,22,24,26,27,29,31,32,33,34]). None reported ABPM acceptance rates by participants or addressed the challenges of extended monitoring in cognitively impaired individuals.

The majority of investigations [7,8,9,10,11,12,13,14,15,16,17,18,19,21,22,23,24,25,27,29,30,31,32,34] (24/28, 85.7%) relied on fixed time periods to define the sleep and wake spans of participants when analyzing the around-the-clock ABPM-derived measures. Only one study [20] (1/28, 3.6%) reported using a sleep diary to determine individual sleep-wake cycles, while three studies [26,28,33] (3/28, 10.7%) did not specify their method for determining them. No or inappropriate consideration of the sleep/wake schedule potentially lead to inaccurate or incorrect BP changes from daytime/awake to nighttime/asleep— key measures to determine dipping status. In addition, the definition of dipping pattern was inconsistent across studies. While several studies [8,9,11,12,16,17,20,23,26,31,34] (11/28, 39.3%) used SBP to define dipping status, some used both SBP and DBP [7,13,14,15,24,25,28] (7/28, 25.0%), or mean arterial pressure [18] (1/28, 3.6%). Nine studies [10,19,21,22,27,29,30,32,33] (32.1%) failed to specify which BP measurement was used for dipping status. The threshold for defining dipping status also varied, with most studies using a 10% decrease in nocturnal mean BP from daytime mean BP.

Control for confounding factors was inconsistent; only 13 studies [8,9,11,17,18,20,23,24,29,30,31,32,34] (46.4%) reported adjustment of potentially influential variables such as age, sex, education level, and comorbidities. Notably, none of the studies controlled for antihypertensive medication timing/administration schedule. Statistical reporting was often incomplete, e.g., only 15 studies [9,11,12,17,19,20,21,22,23,24,29,30,31,32,34] (53.6%) reported effect size, while many solely provided *p*-values.

## 4. Discussion

Our critical analysis of the methodological approaches utilized to assess the relationship between ABPM patterns and cognitive function aimed to identify key limitations and areas for improvement in this important field of research. Our review revealed several significant methodological shortcomings that potentially compromise the reliability and generalizability of current findings. These limitations include a lack of diversity in study populations (with race/ethnicity unreported in 82% of studies), inconsistent ABPM protocols, inadequate control for confounding variables (with only 46.4% of studies adjusting for potential confounders), and a scarcity of longitudinal studies (only 10.7% of studies).

While our analysis entailed the same 28 investigations included in Gavriilaki et al.’s comprehensive 2023 systematic review [5], which represents the most recent synthesis of the literature and focused on quantifying ABPM-cognition associations, it focuses on the differences and shortcomings of their methods, thereby providing a distinctly distinctive and important contribution. Specifically, while Gavriilaki et al.’s meta-analysis demonstrated strong associations between BP patterns and cognitive outcomes, our systematic examination reveals important limitations in how these relationships have been studied. By identifying specific methodological shortcomings—from study design to data analysis—it is our intention to provide a roadmap for strengthening such research endeavors. 

A significant deficiency of these past studies is the widespread lack of reporting of racial and ethnic demographics, with 82% of them failing to specify the racial/ethnic composition of the study populations. Among the few studies that did report this information, there was limited inclusion of Black and Latino participants. This reporting gap makes it impossible to assess the true representation of different racial and ethnic groups in these studies. The lack of demographic reporting and potential underrepresentation are particularly concerning given the well-documented disparities in the prevalence and outcomes of hypertension among these groups. For instance, Lackland (2014) [35] reported that African Americans exhibit a significantly higher prevalence of hypertension compared to other racial groups, with earlier onset and more severe consequences. The limited inclusion of diverse populations in ABPM and cognitive function studies may lead to findings that do not accurately represent the full spectrum of the relationship between BP patterning and cognitive outcomes across different racial and ethnic groups.

Past studies predominantly focused on all-cause dementia or Alzheimer’s disease, and there was only limited investigation of other subtypes of dementia. It’s important to note that this narrow focus is not a result of our own inclusion/exclusion criteria, as we did not restrict the review of studies based on dementia subtype. Given that different subtypes of dementia have distinct pathophysiological mechanisms, it is plausible their relationships with diurnal BP patterning may vary. For example, vascular contributions to frontotemporal dementia have been increasingly recognized [36], suggesting the nature of the ABPM pattern might have unique association with this subtype of dementia. Consideration of different types of dementia will help identify the potential differences in the association between ABPM patterning and specific forms of cognitive decline.

The heterogeneity and limitations of cognitive assessment methods employed across studies significantly hinder our understanding of the nuanced relationship between ABPM patterning and cognitive function. The prevalent use of brief screening tools, such as the MMSE, while practical, may obscure subtle cognitive changes associated with BP variability. These global measures lack the sensitivity to detect domain-specific impairments that could be differentially affected by ABPM patterns. For instance, various cognitive domains may be impacted by BP 24 h variability, but these nuanced effects might go undetected by use of general cognitive screens. Moreover, the reliance on simple cut-off scores to define cognitive impairment potentially obscures the identification of those individuals in preclinical stages of decline, precisely when ABPM patterns might be most informative. To address these shortcomings, future research should employ comprehensive neuropsychological batteries that assess multiple cognitive domains independently, alongside more sensitive measures designed to detect early cognitive changes. This approach would not only provide a more accurate picture of the ABPM-cognition relationship but also help identify subtle impairments that could be crucial for early interventional strategies.

Another methodological concern on several studies is the use of only of wake-time office BP measurements (OBPM), rather than ABPM, to define hypertension in their inclusion/exclusion criteria. This approach may lead to misclassification of hypertensive status, with the risk of biasing study populations and reported findings of studies. The superiority of ABPM over OBPM in predicting health outcomes has been well-established [37,38,39,40,41], supporting that ABPM should be the gold standard for defining hypertension [42].

All of the 28 studies included in our review relied on 24 h ABPM, despite compelling evidence that 48 h ABPM provides much more representative and reliable data. Hermida et al. (2013) [43] demonstrated that 48-h, compared to 24-h, ABPM significantly improves the accuracy of diagnosis and risk for cardiovascular disease events. The first day of ABPM may be slightly biased due to an “ABPM effect”—BP values somewhat higher than actual values during the initial hours of measurement. This effect can persist for up to 9 h and result in an average increase of as much as 7 and 5 mm Hg in SBP and DBP, respectively, during the first 4 h of monitoring [44]. This phenomenon is distinct from white coat hypertension and can lead to misclassification of patients’ dipping status. Importantly, Hermida et al. (2002) [44] found that one-third of patients classified as dippers based on the first 24 h of the 48 h monitoring were reclassified as non-dippers when utilizing all the data of 48 h ABPM assessment. Our review of the 28 studies of the Gavriilaki et al. (2023) publication [5] additionally revealed considerable variability in ABPM sampling rates. Most studies used a sampled BP every 15–30 min during the day and 30–60 min at night. However, the reproducibility of ABPM-derived parameters depends more on the duration than on the frequency of sampling of monitoring [43]. BP means estimated from data sampled every 1–2 h over 48 h were more reproducible than those estimated from data sampled every 20–30 min for only 24 h. This suggests that extending ABPM duration to 48 h, even with a reduced sampling frequency, could provide more reliable and clinically valuable information than the current standard of frequent measurements of just 24-h monitoring. Moreover, this approach also improves tolerance to ABPM, a vital consideration when applied to cognitively impaired persons.

Another critical aspect of the methods of ABPM that warrants attention is the necessity for rigorous data quality control, which encompasses both the elimination of erroneous values and the establishment of thresholds for missing or invalid data to qualify BP profiles as acceptable for analysis. The integrity and reliability of ABPM data are paramount for accurately assessing BP patterns and their relationship to cognitive outcomes. Our review observed heterogeneity in data preprocessing approaches, with 64.3% of studies failing to report any procedures or criteria of data editing. This variability may contribute to inconsistencies in findings across studies. One exemplary study [30] stipulated that >80% of programmed values with no more than 2 h of missing data were required for a BP profile to be considered valid and to ensure a minimum density of correct measurements. Such criteria are especially crucial for nighttime measurements, where a paucity of readings could lead to misclassification of inaccurate estimation of nocturnal BP and thus BP dipping status.

Inconsistencies in dipping definitions across studies present additional challenge. While 39.3% of studies used SBP, 25% used both SBP and DBP, and 3.6% used mean arterial pressure. Notably, 32.1% failed to specify which BP measurement was used to define dipping status. This inconsistency may lead to discrepancies in results across studies. Standardization of dipping definitions is crucial for comparability and meta-analysis of findings. In addition, daily rhythms of behaviors, such as in sleep/wake routine and physical activity, can also affect the quantification of BP dipping. Hermida et al. (2019) [45] emphasize this point, highlighting the distinction between “nighttime” BP and “sleep-time” BP. They argue that daily rhythms in neuroendocrine, endothelial, vasoactive peptide, opioid, and hemodynamic parameters—including renin, angiotensin, and aldosterone—that are primary determinants of the BP 24 h pattern are all affected by or aligned to the 24 h rest/activity cycle. In other words, the timings of these rhythms in terms of clock time can vary between individuals who adhere to different sleep/wake schedules. Reliance on arbitrary fixed clock hours to define the sleep and wake spans of participants, as conventionally practiced in most of the 28 reviewed studies (85.5%), is unlikely to be representative of the individualized rest/activity patterns and constitutes a major shortcoming in the calculation of BP dipping. Thus, a significant limitation of these past studies is the derivation of non-biological relevant “daytime/nighttime” BP rather than biologically meaningful awake and asleep BP. This fixed-time approach leads to misclassification of sleep and wake periods, potentially affecting the reported association between features of the 24 h BP profile and cognitive outcomes. The use of more precise methods to determine individual sleep-wake cycles and sleep/wake status, such as actigraphy or detailed sleep diaries, enables accurate determination of the awake and asleep periods and the calculation of BP means and consequently dipping status to properly assess the relationship between diurnal BP patterning and cognitive function.

Inadequate control for confounding variables was observed in many studies. Factors such as age, sex, education level, body mass index, smoking status, alcohol consumption, and comorbidities of diabetes and cardiovascular disease can significantly influence both BP 24 h patterning and cognitive function [1]. Future studies should consistently control for these variables to isolate the specific relationship between features of the 24 h BP pattern and cognitive outcomes. Additionally, none of the studies reported controlling for the timing/schedule of antihypertensive medication. Given the large amount of existing knowledge that the timing of medication can significantly affect ABPM patterns [46,47,48,49,50,51,52,53,54], this is a crucial factor to consider in future studies. The potential timing effect of medication on the relationship between ABPM patterns and cognitive function remains an important area for investigation.

Another notable limitation observed across most of the reviewed studies was the lack of reporting on ABPM acceptance rates by participants. This omission represents a significant gap in our understanding of the feasibility and acceptability of ABPM, particularly in populations with cognitive impairment or dementia. The successful acceptance and completion of ABPM protocols is crucial for obtaining reliable and representative data; yet, the challenges associated with extended monitoring periods in cognitively vulnerable individuals remain largely unexplored. This paucity of information is particularly concerning given the potential impact of cognitive status on adherence to ABPM protocols. Individuals with dementia may experience increased distress or confusion during monitoring, resulting in a high rate of rejection or incomplete or invalid ABPM profiles. Furthermore, the absence of completion rate data precludes a comprehensive assessment of potential selection bias, as those unable to complete ABPM might systematically differ in their BP profiles or cognitive characteristics than those who are able to do so. Future research should prioritize the explicit reporting of ABPM acceptance and completion rates, stratified by cognitive status where applicable, to elucidate the practical challenges and limitations of applying this method to assess BP in diverse populations. Such data would not only inform the interpretation of study results but also guide the development of more inclusive and adaptable ABPM protocols for individuals across the cognitive spectrum.

Furthermore, several studies relied solely on *p*-values without reporting effect sizes, limiting the interpretation of the clinical significance of findings and hindering comparison across studies. This practice goes against current statistical reporting recommendations [55], making it challenging to assess the magnitude and practical importance of observed associations.

Finally, another significant limitation is the scarcity of longitudinal studies that examined the bidirectional relationship between features of the 24-h BP pattern and cognitive decline. This gap restricts our understanding of how changes over time in ABPM patterns might precede or follow cognitive decline and vice versa. Longitudinal studies are crucial for establishing temporal relationships and inferring causality as well as opportunities for intervention and prevention.

Our study has some limitations. While our analysis focused on studies included in a recent (2023) systematic review, examining additional studies could potentially provide broader insight into the methodological practices in this area of research. However, the limitations and shortcomings of the investigative methods identified through our analysis of the 28 key studies of the 2023 review highlight critical areas requiring improvement in ABPM-cognition research. Future work could extend this methodological analysis to a broader set of studies using an expanded systematic search strategy. The value of our methodological critique lies in its potential to stimulate more rigorous research design that can deliver reliable, generalizable evidence about the complex relationship between hypertension, BP patterns, and cognitive outcomes.

## 5. Conclusions

This critical review has identified several methodological shortcomings in studies investigating the relationship between ABPM-determined BP patterns and cognitive function. The primary deficiencies include inadequate reporting of racial/ethnic demographics, limited investigation of different dementia subtypes, inconsistent definition of dipping status, reliance on fixed time-based rather than actual sleep-wake spans for BP assessment, insufficient control for confounding variables, and predominance of cross-sectional rather than longitudinal study designs. These methodological issues potentially compromise the validity and generalizability of current evidence regarding the ABPM-cognition relationship. Addressing these limitations is essential for establishing a more robust understanding of how diurnal BP rhythms relate to cognitive function and decline across diverse populations and conditions.

## 6. Future Directions

The field of ABPM and cognitive function research requires significant methodological advancement. Key priorities should include: (1) recruitment strategies ensuring diversity in study populations; (2) comprehensive investigation of specific dementia subtypes beyond Alzheimer’s disease; (3) adoption of 48-hour ABPM with individualized sleep-wake determination through actigraphy or detailed sleep diaries; (4) standardization of dipping definitions and data quality control procedures; (5) systematic control and reporting of antihypertensive medication timing; (6) implementation of domain-specific cognitive assessments capable of detecting subtle impairments; and (7) development of well-designed longitudinal studies with repeated measurements. Collaboration among researchers to establish standardized protocols and reporting guidelines would substantially enhance comparability across studies. Such methodological improvements would not only strengthen the evidence base but also inform clinical practice guidelines for BP management aimed at preserving cognitive health and preventing dementia across increasingly diverse aging populations.

## Figures and Tables

**Table 1 clockssleep-07-00011-t001:** Methodological characteristics of ambulatory blood pressure monitoring (ABPM) studies examining cognitive function.

First Author (Year)	ABPM Duration (Sampling Intervals)	Dipping Definition	ABPM Quality Control	Sleep/Wake Classification	Effect Size Calculation	Report of Dropout or Completion%	Control for Confounding Variables	Control for Timing of BP ^a^ Medication
Cani I (2022) [7]	24 h (Not specified)	SBP ^a^ and DBP ^a^	No	Fixed time	No	No	No	No
Chen HF (2013) [13]	24 h (30 min)	SBP ^a^ or DBP ^a^	No	Fixed time	No	No	No	N/A (no medication)
Cicconetti P (2003) [14]	24 h (Day:15 min, Night: 20 min)	SBP ^a^ and DBP ^a^	SBP ^a^ > 260 and <70, DBP ^a^ > 150 and <20 mmHg values excluded	Fixed time	No	No	No	N/A (no medication)
Cicconetti P (2004) [15]	24 h (Day: 15 min, Night: 20 min)	SBP ^a^ and DBP ^a^	No	Fixed time	No	No	No	N/A (no medication)
Daniela M (2023) [8]	24 h (Day: 15 min, Night: 30 min)	SBP ^a^	No	Fixed time	No	No	sex	No
Ghazi L (2020) [9]	24 h (Not specified)	SBP ^a^	Excluded if <14 daytime readings or <6 nighttime readings	Fixed time	Yes (HR ^a^)	No	clinic site, year, age, race, sex, education, marital status, income, smoking, alcohol use, illicit drug use, BMI ^a^, use of antihypertensive medications, history of hypertension, diabetes mellitus, hyperlipidemia, anemia, C-reactive protein, urine protein-creatinine ratio, depression, stroke, GFR ^a^	No
Gregory MA (2016) [16]	24 h (Day: 30 min, Night: 60 min)	SBP ^a^	No	Fixed time	No	93.5% completion	No	No
Guo H (2010) [17]	24 h (Day: 15 min, Night: 30 min)	SBP ^a^	Excluded BP ^a^ readings beyond specified range	Fixed time	Yes (OR ^a^)	No	age, sex, clinic SBP ^a^, hypnotic treatment, type II diabetes, brachial-ankle pulse wave velocity, Apolipoprotein E ε4 allele	N/A (no medication)
Kececi Savan D (2016) [18]	24 h (Not specified)	MAP ^a^	No	Fixed time	No	No	Stratified by sex	No
Kim JE (2009) [19]	24 h (60 min)	Not specified	No	Fixed time	Yes (OR ^a^)	No	No	No
Komori T (2016) [20]	24 h (30 min)	SBP ^a^	<20 valid awake readings and <6 valid sleep readings excluded after	Sleep diary	Yes (OR ^a^)	87% completion	Age, sex	No
Li XF (2017) [21]	24 h (Day: 30 min, Night: 60 min)	Not specified	omitted all presumed erroneous readings	Fixed time	Yes (Correlation)	No	No	No
Mahmoud KS (2014) [22]	24 h (Day: 30 min, Night: 60 min)	Not specified	No	Fixed time	Yes (Correlation)	No	No	No
Ohya Y (2001) [23]	24 h (30 min)	SBP ^a^	Omitted all presumed erroneous readings	Fixed time	Yes (Correlation)	No	age, Barthel Index, hematocrit, previous stroke	N/A (no medication)
Okuno J (2003) [24]	24 h (Day: 30 min, Night: 60 min)	SBP ^a^ and DBP ^a^, separately	No	Fixed time	Yes (OR ^a^)	<1% not completion	age, sex, education level, diabetes mellitus, heart disease, hypercholesterolemia, current alcohol intake, current smoking, benzodiazepine use, BMI ≥ 25, antihypertensive drug use	No
Paganini-Hill A (2019) [25]	24 h (60 min)	SBP ^a^ and DBP ^a^, separately	Omitted all presumed erroneous readings; <6 valid daytime or nighttime readings excluded	Fixed time	No	81.2% completion	No	No
Shim YS (2022) [10]	24 h (Day: 30 min, Night: 60 min)	Not specified	No	Fixed time	Yes (Regression)	No	No	No
Sierra C (2015) [26]	24 h (Not specified)	SBP ^a^	No	Not specified	No	No	No	N/A (no medication)
Suzuki R (2011) [27]	24 h (60 min)	Not specified	No	Fixed time	No	No	No	No
Tadic M (2019) [28]	24 h (20 min)	SBP ^a^ and DBP ^a^, separately	Edited for artifact (no detail)	Not specified	No	No	No	No
Tan X (2021) [11]	24 h (Day: 20 or 30 min, Night: 20 or 60 min)	SBP ^a^	Omitted all presumed erroneous readings	Fixed time	Yes (HR ^a^)	No	BP ^a^ dipping status, age, BMI ^a^, education, daytime SBP ^a^, treatment of hypertension, diabetes, hyperlipidemia, physical activity level, smoking habit, living status	No
Tanaka R (2018) [29]	24 h (Day: 30 min, Night: 60 min)	Not specified	No	Fixed time	Yes (OR ^a^)	97.9% completion	age, sex, Hoehn and Yahr Scale, diabetes, history of stroke, cerebrovascular lesions, orthostatic hypotension	No
White WB (2018) [30]	24 h (Day: 15 min, Night: 30 min)	Not specified	>80% of programmed values; <2 h of missing data required	Fixed time	Regression coefficients	No	age, sex, LDL cholesterol, BMI ^a^	No
Xing Y (2021) [12]	24 h (Day: 30 min, Night: 60 min)	SBP ^a^	No	Fixed time	Yes (Correlation)	71.7% completion	No	No
Yamamoto Y (2002) [34]	24 h (30 min)	SBP ^a,b^	No	Fixed time	Yes (HR ^a^)	No	age and sex	N/A (4-week washout)
Yamamoto Y (2005) [31]	24 h (30 min)	SBP ^a^	No	Fixed time	Yes (OR ^a^)	No	age, sex, PVH ^a^, and nighttime SBP ^a^	N/A (2–4 weeks washout)
Yamamoto Y (2011) [32]	24 h (30 min)	Not specified	No	Fixed time	Yes (OR ^a^)	No	age, sex, 24 h SBP ^a^, estimated GFR ^a^, white matter lesion grade, lacunar infarct grade	N/A (>2 weeks washout)
Yaneva-Sirakova T (2016) [33]	Not specified (Not specified)	Not specified	No	Not specified	No	No	No	No

^a^ Abbreviations: ABPM: ambulatory blood pressure monitoring; BMI: body mass index; BP: blood pressure; DBP: diastolic blood pressure; GFR: Glomerular Filtration Rate; HR: Hazard Ratio; LDL: Low-Density Lipoprotein; MAP: mean arterial pressure; N/A: Not Applicable; OR: Odds Ratio; PVH: Periventricular Hyperintensities; SBP: systolic blood pressure. ^b^ This study, unlike others, considered the threshold definition of dipper/non-dipper status as 5% decline of nocturnal mean BP from daytime mean BP.

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
