# Peer review of "Critical Review of the Methodological Shortcoming of Ambulatory Blood Pressure Monitoring and Cognitive Function Studies"

_2624-5175, 2025, doi:10.3390/clockssleep7010011_

Round 1
Reviewer 1 Report
Comments and Suggestions for Authors
The purpose of this study was to re-investigate a recent systematic review and meta-analysis conducted by Gavriilaki et al. 2023 that examined the relationship between features of Ambulatory Blood Pressure (ABPM) -derived BP patterns and cognitive function across 28 studies involving 7,595 participants. They reported individuals exhibiting normal nocturnal BP dipping had 51% lower risk of cognitive impairment or dementia compared to non-dippers. Furthermore, reverse dippers had up to a 6-fold higher risk of cognitive impairment compared to dippers. These findings highlight the potential prognostic value of features of the BP 24-hour pattern in identifying individuals at elevated risk for cognitive decline.
To evaluate the methodological aspects of these studies, the authors extracted relevant information from each of the reviewed articles using a pre-determined list of relevant elements to capture key aspects of study design and execution that could impact the validity and reliability of findings. The pre-determined list of extracted data/information encompassed study design, sample size, and subject characteristics, duration of ABPM, frequency of BP measurements, definition of BP dipping patterns, criteria for determining validity of ABPM measurements, method of determining daytime wake and nighttime sleep periods, inclusion of follow-up ABPM and cognitive assessments (where applicable), report of sample size/effect size calculation, statement of participant dropout rate, control for confounding factors, and listing or control of the timing of BP medication.
This critical analysis of methodological approaches in the studies of the relationship between ABPM patterns and cognitive function identified key limitations and areas for improvement in this important field of research, like a lack of diversity in study populations, inconsistent ABPM protocols, inadequate control for confounding variables, and a scarcity of longitudinal studies.
The presentation of the investigating design and results is clear and well structured. The result quality is convincing and the shortcomings of the published studies on the relation between ABPM measured blood pressure and cognitive Function are well extracted and focussed. In summary this is as good quality study with an interesting topic that meets today’s requirements for comparative study design and results.
Author Response
Reviewer #2:
The purpose of this study was to re-investigate a recent systematic review and meta-analysis conducted by Gavriilaki et al. 2023 that examined the relationship between features of Ambulatory Blood Pressure (ABPM) -derived BP patterns and cognitive function across 28 studies involving 7,595 participants. They reported individuals exhibiting normal nocturnal BP dipping had 51% lower risk of cognitive impairment or dementia compared to non-dippers. Furthermore, reverse dippers had up to a 6-fold higher risk of cognitive impairment compared to dippers. These findings highlight the potential prognostic value of features of the BP 24-hour pattern in identifying individuals at elevated risk for cognitive decline.
To evaluate the methodological aspects of these studies, the authors extracted relevant information from each of the reviewed articles using a pre-determined list of relevant elements to capture key aspects of study design and execution that could impact the validity and reliability of findings. The pre-determined list of extracted data/information encompassed study design, sample size, and subject characteristics, duration of ABPM, frequency of BP measurements, definition of BP dipping patterns, criteria for determining validity of ABPM measurements, method of determining daytime wake and nighttime sleep periods, inclusion of follow-up ABPM and cognitive assessments (where applicable), report of sample size/effect size calculation, statement of participant dropout rate, control for confounding factors, and listing or control of the timing of BP medication.
This critical analysis of methodological approaches in the studies of the relationship between ABPM patterns and cognitive function identified key limitations and areas for improvement in this important field of research, like a lack of diversity in study populations, inconsistent ABPM protocols, inadequate control for confounding variables, and a scarcity of longitudinal studies.
The presentation of the investigating design and results is clear and well structured. The result quality is convincing and the shortcomings of the published studies on the relation between ABPM measured blood pressure and cognitive Function are well extracted and focussed. In summary this is as good quality study with an interesting topic that meets today’s requirements for comparative study design and results.
Authors’ Response:
We sincerely thank Reviewer #2 for their thorough and positive evaluation of our manuscript. We appreciate their recognition of the clear presentation, convincing quality, and effective analysis of methodological shortcomings in published studies examining ABPM and cognitive function. Their detailed understanding of our study's methodology and findings, as reflected in their comprehensive summary, validates our approach to identifying key areas for improvement in this important field of research.
No specific revisions were requested by this reviewer. We thank them for their careful review and supportive assessment.
Reviewer 2 Report
Comments and Suggestions for Authors
In this manuscript, with the assistance of an artificial intelligence language model Claude AI (Claude 3.5 Sonnet, Anthropic), Haghayegh et al. reanalyzed the 28 studies collected in the systematic review on the relationship between ABPM patterns with cognitive function and risk of dementia, conducted by Gavriilaki et al., and revealed several significant limitations in the current studies of ABPM and cognition in terms of study design, sample characteristics, ABPM protocol, cognitive assessment, and data analysis. The authors also provided seven well-thought-out suggestions for the field. Both shortcomings and suggestions are clearly elucidated and useful. I have the following concerns and suggestions.
1. The review is based solely on 28 previously selected studies, which could bias the overall assessment of the relationship between ABPM patterns and cognitive function. The authors should consider obtaining additional relevant studies or emphasize the limitations of this review.
2. As the relevant information from 28 studies was reviewed previously, both Tables 1 and 2 should be significantly curtailed. The authors should consider putting both or at least one into the supplementary file.
3. It may be intuitive that the current studies lack diversity in study populations, particularly the underrepresentation of Blacks and Latinos. However, among 28 studies, 23 studies did not specify the race of the participants, which undermines the argument.
4. While the authors qualitatively describe the heterogeneity and limitations of the 28 studies, they should consider using more quantitative analyses to strengthen the arguments. For instance, the proportion of studies affected by each method should be calculated more precisely, or a meta-regression should be conducted to investigate the effect of different methodological factors on the reported associations.
5. "n Hu", the first name of the last author may be misspelled.
Author Response
In this manuscript, with the assistance of an artificial intelligence language model Claude AI (Claude 3.5 Sonnet, Anthropic), Haghayegh et al. reanalyzed the 28 studies collected in the systematic review on the relationship between ABPM patterns with cognitive function and risk of dementia, conducted by Gavriilaki et al., and revealed several significant limitations in the current studies of ABPM and cognition in terms of study design, sample characteristics, ABPM protocol, cognitive assessment, and data analysis. The authors also provided seven well-thought-out suggestions for the field. Both shortcomings and suggestions are clearly elucidated and useful. I have the following concerns and suggestions.
Authors’ Response:
We thank the reviewer for their careful review and constructive feedback. We appreciate their suggestions for improvement.
Comment #1: The review is based solely on 28 previously selected studies, which could bias the overall assessment of the relationship between ABPM patterns and cognitive function. The authors should consider obtaining additional relevant studies or emphasize the limitations of this review.
Authors’ Response: We appreciate this important point about the potential limitations of analyzing only the 28 studies from Gavriilaki et al.'s systematic review. Our focus on these studies was intentional, as they represent the most comprehensive and recent (2023) collection of research examining ABPM-cognition relationships. Our aim was to conduct an in-depth methodological analysis of studies previously determined to meet inclusion criteria for examining these relationships, in contrast to Gavriilaki et al.'s focus on synthesizing outcome data and quantifying associations between ABPM patterns and cognitive function. However, we agree this should be explicitly addressed. We added a paragraph to the Discussion acknowledging this limitation and noting that future work could extend this methodological analysis to a broader set of studies. We specifically added these two paragraphs:
“While our analysis focused on the 28 studies included in Gavriilaki et al.'s comprehensive 2023 systematic review [5], which represents the most recent synthesis of this literature and focused on quantifying ABPM-cognition associations, our methodological approach provides a distinct contribution. While their meta-analysis demonstrated strong associations between BP patterns and cognitive outcomes, our systematic examination of research methods reveals important limitations in how these relationships have been studied. By identifying specific methodological shortcomings - from study design to data analysis - our work provides a roadmap for strengthening future research in this field. The methodological improvements we recommend, such as extending ABPM duration to 48 hours, standardizing cognitive assessments, and better controlling for confounding variables, will help produce more reliable and generalizable evidence about the relationship between BP patterns and cognitive function. This enhanced methodological rigor is essential for developing evidence-based guidelines for BP monitoring and control to prevent cognitive decline.”
“Our study has some limitations. While our analysis focused on studies included in a recent (2023) systematic review, examining additional studies could potentially provide broader insights into methodological practices in this field. However, the methodological limitations identified through our analysis of these key studies highlight critical areas requiring improvement in ABPM-cognition research. Future work could extend this methodological analysis to a broader set of studies using an expanded systematic search strategy.”
Comment #2: As the relevant information from 28 studies was reviewed previously, both Tables 1 and 2 should be significantly curtailed. The authors should consider putting both or at least one into the supplementary file.
Authors’ Response: Thank you for this suggestion about the tables. As suggested, we have moved Table 1 to the Supplementary Materials as this information was previously presented in Gavriilaki et al.'s review. However, we have retained Table 2 in the main manuscript as it contains the core methodological data that forms the basis of our analysis and presents information not evaluated in the previous systematic review. This table provides critical details about ABPM protocols, data quality control, sleep/wake classifications, and other methodological aspects that are central to our paper's contribution.
Comment #3: It may be intuitive that the current studies lack diversity in study populations, particularly the underrepresentation of Blacks and Latinos. However, among 28 studies, 23 studies did not specify the race of the participants, which undermines the argument.
Authors’ Response: We appreciate this keen observation about our discussion of population diversity. The reviewer is correct that race/ethnicity was unreported in 82% (23/28) of studies, which affects our original framing. We have revised our manuscript to emphasize that this widespread lack of reporting of racial/ethnic demographics is itself a significant methodological limitation that needs to be addressed in future research. This reporting gap makes it impossible to assess the true representation of different racial and ethnic groups in these studies, which is particularly concerning given the well-documented disparities in hypertension prevalence and outcomes across different populations. We specifically added these sections:
“The study populations had limited demographic reporting, with race/ethnicity unreported in 82% (23/28) of studies. Among studies that did report racial/ethnic demographics, only 2 studies [9,30] included Black or Latino participants.”
“A significant concern in included studies is the widespread lack of reporting of racial and ethnic demographics, with 82% of studies not specifying the racial/ethnic composition of their study populations. Among the few studies that did report this information, there was limited inclusion of Black and Latino participants. This reporting gap makes it impossible to assess the true representation of different racial and ethnic groups in these studies. The lack of demographic reporting and potential underrepresentation are particularly concerning given the well-documented disparities in the prevalence and outcomes of hypertension among these groups.”
Comment #4: While the authors qualitatively describe the heterogeneity and limitations of the 28 studies, they should consider using more quantitative analyses to strengthen the arguments. For instance, the proportion of studies affected by each method should be calculated more precisely, or a meta-regression should be conducted to investigate the effect of different methodological factors on the reported associations.
Authors’ Response: We thank the reviewer for this valuable suggestion to strengthen our analysis with more quantitative details. We have thoroughly revised the manuscript to include precise calculations of the proportion of studies affected by each methodological limitation and have enhanced our presentation of these statistics.
We appreciate the suggestion of meta-regression analysis and we carefully considered it, but several methodological factors make this approach challenging for our type of review. Meta-regression is specifically designed to examine how study characteristics influence effect sizes or outcomes, while our paper focuses on evaluating research methods rather than synthesizing outcomes (which was already addressed in Gavriilaki et al.'s meta-analysis). A fundamental requirement for meta-regression is the consistent reporting of effect sizes across studies. However, in our sample, only 53.6% of studies reported effect sizes, and these were presented using different measures that cannot be meaningfully combined. Furthermore, the methodological characteristics we evaluated were often incompletely reported or presented in non-standardized ways across studies, preventing meaningful quantitative synthesis through meta-regression. Many of our key findings relate to methodological limitations such as inadequate reporting of protocols, lack of quality control measures, and inconsistent definitions of key variables - aspects that cannot be appropriately analyzed through meta-regression techniques. Instead, we have strengthened our analysis by providing comprehensive quantitative details about the prevalence of various methodological characteristics to support our arguments.
Comment #5: "n Hu", the first name of the last author may be misspelled.
Authors’ Response: Thank you for catching this typographical error. We have corrected the spelling of the last author's name from "n Hu" to "Kun Hu" in the author list.
Reviewer 3 Report
Comments and Suggestions for Authors
This manuscript provided a comprehensive review on recent studies on the relationship between blood pressure and cognitive function. The authors took great effort to point out the shortcomings in the methodologies adopted by those previous studies. Those include ethical composition of the population used, the use of 24-hr ABPM instead of 48-hr ABPM, and the inadequacy in rigorous data quality control. Overall, the information in this manuscript is of value to future studies.
Author Response
This manuscript provided a comprehensive review on recent studies on the relationship between blood pressure and cognitive function. The authors took great effort to point out the shortcomings in the methodologies adopted by those previous studies. Those include ethical composition of the population used, the use of 24-hr ABPM instead of 48-hr ABPM, and the inadequacy in rigorous data quality control. Overall, the information in this manuscript is of value to future studies.
Authors’ Response:
We thank Reviewer #3 for their positive assessment of our manuscript. We appreciate their recognition of our comprehensive analysis and their acknowledgment of how we identified key methodological limitations in previous studies, including issues with population diversity, ABPM duration, and data quality control. We are pleased that they find our work valuable for informing future research. No specific revisions were requested by this reviewer.
Round 2
Reviewer 2 Report
Comments and Suggestions for Authors
Thanks for addressing my comments and concerns.